# A Field-Programmable Gate Array-Based Adaptive Sleep Posture Analysis Accelerator for Real-Time Monitoring

**DOI:** 10.3390/s24227104

**Published:** 2024-11-05

**Authors:** Mangali Sravanthi, Sravan Kumar Gunturi, Mangali Chinna Chinnaiah, Siew-Kei Lam, G. Divya Vani, Mudasar Basha, Narambhatla Janardhan, Dodde Hari Krishna, Sanjay Dubey

**Affiliations:** 1Department of Electronics and Communication Engineering, Koneru Lakshmaiah Education Foundation, Aziz Nagar, Hyderabad 500075, Telangana, India or sravanthi.engg@gmail.com (M.S.); sravankumar.gunturi@gmail.com (S.K.G.); 2Department of Electronics and Communication Engineering, Malla Reddy Institute of Engineering and Technology, Maisammaguda, Hyderabad 500014, Telangana, India; 3Department of Electronics and Communications Engineering, B. V. Raju Institute of Technology, Medak (Dist), Narsapur 502313, Telangana, India; divyavani.g@bvrit.ac.in (G.D.V.); mudasar.basha@bvrit.ac.in (M.B.); harikrishna.dodde@bvrit.ac.in (D.H.K.); sanjay.dubey@bvrit.ac.in (S.D.); 4College of Computing and Data Science (CCDS), Nanyang Technological University, Singapore 639798, Singapore; siewkei_lam@pmail.ntu.edu.sg; 5Department of Mechanical Engineering, Chaitanya Bharati Institute of Technology, Gandipet, Hyderabad 500075, Telangana, India; njanardhan_mech@cbit.ac.in

**Keywords:** sleep posture recognition, adaptive posture analysis, FPGA, sensor fusion

## Abstract

This research presents a sleep posture monitoring system designed to assist the elderly and patient attendees. Monitoring sleep posture in real time is challenging, and this approach introduces hardware-based edge computation methods. Initially, we detected the postures using minimally optimized sensing modules and fusion techniques. This was achieved based on subject (human) data at standard and adaptive levels using posture-learning processing elements (PEs). Intermittent posture evaluation was performed with respect to static and adaptive PEs. The final stage was accomplished using the learned subject posture data versus the real-time posture data using posture classification. An FPGA-based Hierarchical Binary Classifier (HBC) algorithm was developed to learn and evaluate sleep posture in real time. The IoT and display devices were used to communicate the monitored posture to attendant/support services. Posture learning and analysis were developed using customized, reconfigurable VLSI architectures for sensor fusion, control, and communication modules in static and adaptive scenarios. The proposed algorithms were coded in Verilog HDL, simulated, and synthesized using VIVADO 2017.3. A Zed Board-based field-programmable gate array (FPGA) Xilinx board was used for experimental validation.

## 1. Introduction

The human life cycle and health are linked to sleep duration and posture. Over the last three decades, researchers have conducted sleep analysis and monitoring. According to medical reports and history, the impacts of improper sleep on daily life include musculoskeletal strain, respiratory issues, circulation problems, reduced sleep quality, and digestive issues. Indirect poor sleep affects human behavior and daily activities. Research has shown that 9% to 38% of the general population is affected by sleep apnea [1]; in the future, this is expected to increase. Sleep posture recognition and analysis are crucial for researchers and medical systems when recommending various medications and other equivalent systems for better sleep in patients as well as the general population. The American Sleep Disorders Association and Sleep Research Society have been investigating the impact of sleep disorders on human activities for the last four decades [2,3].

To date, sleep posture analysis has faced various challenges, including data acquisition, in assisting patients and individuals. Data acquisition has been performed by researchers using wearable and non-wearable pressure and non-contact sensing devices [4,5]. Piezoresistive arrays or pressure-sensor-based bedsheets and beds are utilized for data acquisition for posture analysis [5,6]. A few challenges are raised with these methods; the data acquisition error percentage is higher in this case, and a large amount of data are required to compute the posture. Other researchers have conducted posture analysis using wearable devices; this approach has been used in initial learning and regular patient monitoring [7,8]. Recent advancements in sleep posture data acquisition have been achieved using non-contact methods, such as radar or ultrasonic sensor arrays [5] and vision approaches. Wearable sensors such as 3-axis accelerometers [9] and thermostats, as well as electromyography (EMG), electroencephalography (EEG), and photoplethysmography (PPG) devices [10], have been utilized for sleep posture data capture. These wearable sensors are integrated into Fit-Bit modules or smart watches. Similarly, non-wearable technology and sensors, such as multimodal sensor fusion [11] and smart phones [12], have been utilized for sleep posture analysis by various researchers. Selection of the sensor type is a challenge in sleep posture analysis.

Different sensor data have been processed using data processing analysis techniques, such as data mining and classifiers. Computation methods play a vital role in sleep posture data analysis. Classifiers such as random forests and binary-type decision trees, as well as supervised classifiers such as hidden Markov models (HMMs), support vector machines (SVMs), and k-nearest neighbors (kNNs) [13], have been employed. Classifiers are used for learning, and real-time feature matching can be utilized to determine sleep posture. In this regard, researcher-driven deep learning methods have been adopted in sleep research, such as recurrent NNs (RNNs), long short-term memory (LSTM) networks [6], convolutional NNs (CNNs), and generative adversarial NNs (GNNs) [14]. In this process, computing devices such as microcontrollers, GPUs, and CPUs, as well as cloud computing, are essential for sleep posture data acquisition, data processing with classifiers, and posture accomplishment [15,16]. Research studies addressing sleep posture require low-power-consuming devices and effective computation for analysis during learning and real-time implementation in static and adaptive states. Edge computing devices, such as FPGAs, are essential for real-time sleep posture systems at present and in the future, making it challenging to provide complete solutions.

The proposed hardware-efficient methods provide sleep posture analysis for present and future usage in real-time implementations. The FPGA-based adaptive sleep posture analysis accelerator has three novel embedded methods:An FPGA-based learning algorithm for acquiring data regarding the standard and adaptive conditions of sleep postures. Real-time adaptive learning was developed for various subjects (humans).An FPGA-based hierarchical binary classifier (HBC) algorithm was developed for the classification of sensor fusion data in the learning and analysis stages for event-driven conditions.Hardware-based sleep posture analysis is the next stage of the proposed method. The FPGA-based solution for sleep posture analysis is the first of its kind for adaptive-based event conditions.

This paper is structured as follows. This section presents the background and motivation for sleep posture analysis research using FPGA implementation. In Section 2, the details of the proposed methodology are presented with theoretical and hardware schemes. The proposed method was validated, and the results are presented in Section 3 in the form of synthesis, power consumption, and experimental details with comparison. The final section concludes the study with future perspectives.

## 2. Hardware-Based Algorithms

Sleep posture analysis was executed in two stages as per the proposed method. Hardware-based algorithms were first developed for sleep posture analysis, and secondly, hardware schemes for analysis were explored. See Table 1.

### 2.1. Hardware-Based Algorithm for Sleep Posture Analysis

Figure 1 presents an overall flowchart of the proposed hardware-based sleep analysis. The initial conditions are as follows: The algorithm checks whether there is a subject on the bed. Next, it determines whether data have previously been recorded for this subject’s profile. If the subject is new to the sensor radar, it starts learning all their postures; such an adaptive process not only relies on near-sensor fusion data. If the subject is recognized, their posture is classified using a hardware-based hierarchical binary classifier (HBC) algorithm. The HBC provides the along class, and the subject is either in a static or adaptive state. In a static pose, the subject’s posture is recorded. If the subject switches from one pose to another, this is recorded along with the time spent in the respective postures.

The sleep posture details are presented in Figure 2 and are broadly classified into four groups: supine (SP), left (LP), right (RP), and frog posture (FP). Posture details are learned by an edge computing device using sensor fusion data. These posture data are learned at the sleep-posture-learning stage and are utilized in the sleep posture analysis in the classifier stage, defining the type of posture.

#### 2.1.1. Hardware-Based Algorithm for Sleep Posture Learning

This section addresses sleep posture learning. Algorithm 1 represents the pseudocode of the sleep posture learning in versatile scenarios.
**Algorithm 1:** Pseudocode for hardware-based sleep posture learning1.       Initialize sensory distance into sensor fusion data (SF)2.       always @ (posedge clk) begin3.              {sleep_posture_analysis, learn_new_posture, adapt_sf_data, retry} = 0.4.       Case (state)5.       INIT: Next State = WAIT_PIR.6.       WAIT_PIR: Next_State = (PIR == 1)? CHECK_DATA_RANGE: WAIT_PIR.7.       CHECK_DATA_RANGE:8.       Next_state= (subject data == range data)? CHECK_POSE: ADAPT9.       CHECK_POSTURE:10.             next_state = (posture_HBC)? ANALYSIS: Learn.11.     ANALYSIS: {sleep_posture_analysis = 1, next_state = WAIT_PIR};12.     LEARN: {learn_new_posture = 1, next_state = WAIT_PIR};13.     ADAPT: {adapt_sf_data = 1, next_state = WAIT_PIR};14.     RETRY: next_state = (PIR == 1)? CHECK_DATA_RANGE: RETRY.15.     default: next_state = INIT.16.                   end case, End.

Algorithm 1 describes the pseudocode for hardware-based sleep posture learning. In step 1 (line 1), the sensors are initialized; the data in the sensor fusion from the sleep posture details are presented in Figure 2. The flowchart of Algorithm 1 is shown in Figure 3. The system clock synchronizes the sensor fusion data and enables the sleep posture to be learned with a continuous, adaptive approach (lines 2 and 3). The PIR sensor data are utilized to determine whether there is a human on the bed (line 6). Sensor fusion data are classified into two forms, standard posture and time-invariant data, which are considered adaptive sleep postures (line 8). Standard posture information related to existing subject postures is then determined (line 10). The proposed method learns any new postures and registers them using the hierarchical binary classifier (HBC) (line 12). Sleep postures are analyzed using the HBC method (line 11). The subject’s movement from one posture to another is evaluated as an adaptive posture (line 13). Adaptive posture evaluation is essential for assisting the subject in their struggle with pain or other issues as it alerts attendants to the medical assistance system. This is a continuous process in the estimation of sleep postures (lines 14 and 15).

#### 2.1.2. Hardware-Based Hierarchical Binary Classifier Algorithm for Sleep Posture

This subsection presents the sensor fusion data classified using the hierarchical binary classifier (HBC) algorithm. Each sleep posture is learned as described in Section 2.1.1 and classified with respect to sleep posture, as shown below.

Figure 4 presents the sensor fusion data captured for the left lateral posture (LLP) and supine posture (SP). Table 2 lists the details of the sensor fusion data for sleep posture analysis. Figure 4 and Table 2 provide basic information for the execution of the hierarchical binary classifier (HBC) algorithm.

Algorithm 2 and the flow chart in Figure 5 provide details of the hardware-based HBC algorithm for sleep posture learning and analysis. Data from the six pairs of sensors are captured from the positions of the human limbs, abdomen, and head. Sensor fusion data initialization and triggered operations are performed in lines 1 and 2, respectively. The HBC classifies data into three positions: lower limb, abdomen, and head (lines 3–14).
**Algorithm 2:** Pseudocode for hardware-based hierarchical binary classifier algorithm1.       Initialize sensory distance into sensor fusion data (SF)2.       always @ (posedge clk) begin {H_R, H_L, A_R, A_L, R_L & L_L} = 0;3.       Case (Posture Data)4.              Posture: (R_L & L_L =2′b00)? Frog posture: Posture _1.5.              Posture _1: (R_L & L_L =2′b11)? Supine posture: Posture _2.6.              Posture _2: (R_L & L_L =2′b01)? Right posture: Left posture.7.         Case (Right posture)8.              Right_1: (A_R & A_L=2′b11)? Right Foetus: Right_2.9.              Right_2: ((A_R & A_L=2′b01) && (H_R & H_L=2′b11))? LY: RLP.10.       Case (Left posture)11.     Left_1: (A_R & A_L=2′b11)? Left Foetus: Left_2.12.     Left_2: ((A_R & A_L=2′b10) && (H_R & H_L=2′b11))? RY: LLP.13.     Default {H_R, H_L, A_R, A_L, R_L & L_L} = 0.end case, End.

Early classification is performed using the lower limb position when the right and left lower limbs (R_L and L_L) are equal to the two-bit information 2′b00. When the limbs are detected as not positioned at the lower end of the bed, the subject’s pose is determined to be a frog posture (line 4). If this condition fails, then the classifier switches to the next state. When the lower limbs are 2′b11, the supine posture is recorded (line 5); otherwise, the process switches to the next state. The classifier utilizes sensor data for the entire lower limbs to establish the right (lines 7 to 9) or left (lines 10 to 12) side postures. When the abdomen sensor data (A_R and A_L) are equal to 2′b11, they are recorded as the right fetal position (line 8), similar to the left fetal position mentioned in line 11. When lower limb and abdomen sensor fusion data cannot be classified, the HBC depends on the head position on the right or left side (H_R and H_L). The final stage of the hierarchy classifies the postures of the left yearner (LY) and right lateral posture (RLP) (line 9), and similar line postures are classified as the right yearner (RY) or left lateral posture (RLP) (line 12) based on the hierarchy conditions. Default conditions are considered to indicate that the postures are not available or improper (line 13). The adaptive approach considers improper posture evaluation for learning and analysis using time-of-flight (ToF) methods.

#### 2.1.3. Hardware-Based Adaptive Sleep Posture Analysis Algorithm

The proposed hardware-based adaptive algorithm for sleep posture analysis under time-varying event conditions is presented in this subsection. Adaptive sleep posture analysis provides accurate information regarding the status of the patients/elderly individual’s motion and sleep conditions. This method depends on the selection of the sensor, which provides the time-of-flight-based sensing, time-stamped data, and time-dependent features and drives the detection of human motion on the bed. Few researchers have used hidden Markov models for motion detection [18,19]. The proposed hardware-based algorithm was developed using a binary-based adaptive threshold for digitizing the motion action in sleep postures. A human transitioning from a left lateral posture (LLP) to a supine posture (SP) is shown in Figure 3.

The sensor captures and calculates the distance to the object or human based on the echo signal between the sensor and human. The formula used is as follows:(1)$$Distance=\frac{Speed\;of\;Signal\times Time\;of\;Flight}{2}$$

Algorithm 3 and Figure 6 present the adaptive sleep posture evaluation method. In this algorithm, the time-variant data are the main concern with respect to posture. This algorithm operates concurrently to evaluate the timing of static and dynamic human postures with respect to other postures. Once the human subject is identified through the PIR sensor using sensor fusion data, their pose is determined. During sleep, patients in pain change from one posture to another. In this context, postural dynamics are essential for estimating patient challenges. Until the pose changes, it is considered to be the same posture as that mentioned in line 4. When posture changes are not observed, the time duration is evaluated using the counter count_1 (lines 9 to 10). If the pose is associated with a time variable, it is evaluated in lines 6–8. The count parameter is deployed to calculate the timing of the dynamic posture variants (line 7). The adaptive posture is defined as a two-level posture, and the process reverts to the original posture, identified as per line 1. Otherwise, the posture is evaluated using the HBC, and the learning time is recorded. This timing is useful, and abrupt changes in the posture time will improve the adaptiveness of the method.
**Algorithm 3:** Pseudocode of adaptive-based algorithm for sleep posture analysis1.       Sensor fusion data & Present posture, count_1 = 8′d0, count = 8′d0.2.       If (PIR)3.       Case (Pose)4.       Pose 1: (Pose current = Pose past)? Same posture: Pose 2.5.       Pose 2: (Pose_T_current = Pose_T_past)? Pose 6: Pose 3.6.              Case (Pose 3)7.              Pose 4: (Pose_ static = Poses)? count: count <= count + 1.8.              Pose 5: (count > count_step1)? HBC: Pose 1.9.              Case (Pose 6)10.            Pose 7: (Pose_ static = Pose current)? count_1: count_1 <= count_1 + 1.11.       end case, End.

## 3. Hardware Schemes for Sleep Posture Analysis

This section outlines hardware schemes that are equivalent to the hardware-based algorithm proposed in Section 2. The hardware schemes for the sleep posture analysis are given in the following. Initially, the hardware accelerator, learning, HBC, and adaptive posture evaluation-based hardware schemes are presented.

### 3.1. Hardware Accelerator for Sleep Posture Analysis

Figure 7 presents the hardware accelerator for the analysis at the learning and evaluation stages. The hardware accelerator is integrated with sensors as represented in orange colours, a Wi-Fi module, and a display unit is represented with red colour line in the figure. the data bus is represented with dark and light blue colour. The overall architecture is controlled using the proposed algorithm with a control unit as represented with yellow colour data bus. The control unit operates and synchronizes the overall system. Passive infrared motion (PIR) sensors enable the system to capture the next level of data. In the absence of a human on the bed, PIR provides logic ‘0’, and the accelerator continues in the sleep condition until PIR = ‘1’. Six pairs of ultrasonic sensors were integrated to capture sleep data. The sensors were concurrently triggered by a control unit. Every 20 bits of sensory data are shared with the distance converter and the fusion module. The control unit initially shares the data with the static posture-learning PE module; if any time-variant data are observed, it switches to an adaptive posture-learning module using a demultiplexer (DEMUX). Posture learning is performed by comparing the sensor fusion data with the reference subject data.

The learning posture utilizes an HBC PE and communicates using 8-bit data. Postures that are not part of the reference model or the time-variant-type posture are learned using the adaptive posture-learning PE. If the pose is within the range of the reference poses, the learning stage enables the posture evaluation of the PE. The posture evaluation PE interfaces with the posture HB classifier module for evaluation. The defined posture is processed as an output through the execution unit (EU). The output is presented in the form of messages to the attendants/service team through the Wi-Fi module. These messages are displayed on the FPGA-based, seven-segment LCD display. This novel approach is used to adaptively analyze variations in the pose of the patient/elderly subject. In the learning stage, if the posture is not a standard pose, it is recorded as an adaptive posture of the subject. Under certain conditions, the subject performs the posture with respect to the time variable, and the adaptive posture evaluation PE registers the posture and compares it to the time-variant data between postures, and this is fed to the output.

### 3.2. Hardware Schemes for Sleep-Posture-Based Learning

The proposed equivalent internal architecture of Algorithm 1, referred to as the integration of the generic and adaptive posture-learning PEs, is presented in Figure 8. The architecture, enabled by the PIR sensor and ultrasonic sensor fusion distance data, has a FIFO memory structure. Prior postures are analyzed and estimated via either generic posture learning or adaptive learning using the posture-matching module. The posture-matching module was developed with three stages of FIFO, which are stored in FIFO_N-2, FIFO_N-1, and FIFO_N. Each FIFO dimension is an array of six pairs of sensors {H_R, H_L, A_R, A_L, R_L, and L_L}, each 20 bits in size. The shift encoder shifts the FIFO data from N-2 to N and performs in line with the concurrent barrel-shifter-based approach.

The FIFO data are stored for every 1/6 s and compared using the ‘comp’ module. L_L sensor-based arrays of N-2, N-1, and N are compared at the L_L comparator at the same distance. H_R to L_L are encoded, and when 6′b11111 is presented in three iterations, new generic learning is enabled in the posture-learning PE. Otherwise, adaptive posture learning is enabled in the PE. The posture-learning PE consists of an internal control module and FIFO posture learning. The internal control shares the present status with the control unit, as presented in Figure 4 with different colours. FIFO posture learning is performed for the standard postures of each subject, as presented in Table 2, from right yearner (RY) to the frog posture (FP). Eight standard postures are included. A maximum of eight subjects are recorded, and the learning methods replace unused subject data with new subject data with control unit permissions. The authors attempted to optimize memory using time-variant-based learning. This method is optimized for storing datasets, as the execution of the real-time interference stage faces memory issues. In this regard, time-variant-based learning is called adaptive learning, which is employed in two scenarios: past subjects with a new posture and new subject postures. The adaptive posture-learning PE is activated as per the posture-matching module; it stores New_1 to New_8. H_R to L_L sensor data are stored as new, standard subject postures. It is also able to memorize new postures for a new subject, which are labeled in the posture-learning PE. A known subject with a new posture is allocated a new label beside the FP posture through the FIFO posture learning. This was a novel attempt, at the inference level, to avoid dataset memory issues. Both learning methods are regularly interfaced with the learning posture classifier PE for posture estimation, as shown in Figure 9.

### 3.3. Hardware Schemes of Hierarchical Binary Classifier for Sleep Posture

Figure 9 presents the internal hardware schemes of the HBC for sleep posture and signal information is represented in different colours. The proposed binary classifier was developed in line with the [20,21] binary search tree. The binary search tree classifier was organized with the center weights of the tree; in the proposed approach, it is heuristic, with a hierarchical binary classifier.

The HBC is embedded with a 20-bit FIFO structure, (sensor distance) × 6 (H_R, H_L, A_R, A_L, R_L and L_L) × 1, and receives the data from the learning or posture evaluation modules, the hierarchical classifier logic, classifier logic control, right posture classifier logic, and left posture classifier logic. As per the binary search tree, R_L and L_L enable 2-bit information to act as a selector in the hierarchical classifier logic. The frog posture (FP) and supine posture (SP) were selected as 2′b00 and 2′b11 and 2′b01 and 2′b10 to enable the right and left posture classifiers. As per the binary hierarchy, with A_R and A_L data, the classifier logic control defines the right and left postures in detail. A_R and A_L as 2′b11 and R_L and L_L as 2′b10 define the right fetal (RF) postures and enable the selection of the right lateral posture (RLP). The next level of hierarchy is defined with the top H_R and H_L bits for the selection of the left postures, such as the lateral, fetal, and yearner postures. Overall, data from the six pairs of sensors (6′b111101) define the left fetal position using the left posture classifier logic. Similarly, lines 6′b010101 classify the left lateral posture (LLP) and 6′b111010 the left yearner (LY) posture. Adaptive learning regularly performs subclassification in the HBC according to new requirements. The posture modules are interfaced with the HBC encoder and with the posture evaluation and learning modules.

### 3.4. Hardware Schemes for Adaptive Posture Evaluation

Algorithm 3 presents hardware-based adaptive posture evaluation using a time-variant approach. The sensor’s distance varies with time and is considered adaptive in sleep posture evaluation. Figure 10 presents the internal hardware schemes for adaptive posture evaluation with different colour lines for data information.

The adaptive posture evaluation is integrated with an adaptive period logic controller and adaptive period calculator. At the logic controller, data originating from the sensor distance bus are utilized to ascertain the past and current poses. When registered concurrently, once the pose is initiated, the respective counters are triggered until the pose changes. These counters operate with a device clock frequency of 100 MHz and are synchronized with the AXI lite protocol. The Pose _ T _ Current and Pose _ T _ past counters are counts utilized in the event of posture registration and reach the threshold for posture change. If both the count and pose are the same, the posture is recognized as the same as that classified with the HBC. Otherwise, the adaptive period calculator of time-variant sleep posture data is enabled. The Pose _ T _ Current module of the period calculator registers data for every 1/6 s and determines the difference using the 2’s complement adder method. The duration from the current posture to the adjacent posture time is considered the adjacent pose period. Once both the adaptive and adjacent pose periods are equal, the HBC evaluates the posture-to-posture time. The adaptive period module iterates until the adjacent pose periods match.

## 4. Results

The proposed FPGA-based accelerator for determining sleep posture under generic and adaptive conditions is presented in this section. The results are compared in terms of FPGA resource utilization in implementation, along with power consumption based on the hardware schemes presented in Figure 7, Figure 8, Figure 9 and Figure 10. The proposed approach was validated through real-time experiments with six ultrasonic sensor pairs and a ZedBoard family field-programmable gate array (FPGA).

### 4.1. Resource Utilization

The proposed approach is the first of its kind to use the FPGA-based accelerator for non-contact sleep posture analysis. The hardware schemes were coded using Verilog HDL, and the Xilinx simulator was used for their functional verification. Vivado tools 17.3 version was utilized for HDL synthesis for bit generation. Xilinx tools and the FPGA were procured from the Xilinx university program.

Xilinx (San Jose, CA, USA) produced the Xilinx Zynq XC7Z020-1CSG484 ZedBoard, which features approximately 85,000 programmable logic cells. The device incorporates look-up tables (LUTs) and flip-flops for executing logic operations and short-term memory storage. The board’s BRAM, accessible via AXI lite, comprises over 140 blocks of 36 kb each (totaling 4.9 Mb), which are used to store sensor fusion and intermediate data. In the proposed design, BRAM is primarily utilized in FIFO. The board also includes about 220 DSP slices (18 × 25 MACCs) for handling data transfer and other computational tasks. Table 3 illustrates how these resources are employed in the proposed approach.

Table 3 presents the device’s utilization of the sleep posture analysis accelerator. FPGA-based accelerators provide fast computing and low power consumption [22,23,24]. The total device utilization for the proposed approach in the form of look-up tables (LUTs), block RAM (BRAM), and digital signal processing (DSP) slices was 46% (24,472), 51% (72), and 44% (96), respectively.

Figure 11 presents a quantitative analysis of the resources consumed by the device in the interfacing module, which are 26% for LUT, 11% for BRAM, and 10% for the DSP slices. Similarly, other modules include the sleep-posture-based-learning PE (18%, 9%, and 7%), hierarchical binary classifier (HBC) PE (16%, 6%, and 6%), adaptive posture evaluation PE (21%, 16%, and 11%), control unit and PWDC sensor fusion (11%, 6%, and 6%), and execution module and display (8%, 4%, and 4%). It is observed from device utilization that the interfacing modules and adaptive posture evaluation PE consume more resources. The interface is embedded with UART using AXI lite to establish and communicate sleep postures to other devices using Wi-Fi ESP8266 external devices. The resource consumption of BRAM is approximately 51%, and the sensor fusion distance is passed in a few stages through FIFO. AXI-based FIFO and an IP core are utilized as part of the programmable logic (PL) of the ZedBoard FPGA.

Figure 12 shows the power consumption of the device when computing a reconfigurable device (FPGA). Overall, the static device power consumption as per the Xilinx power estimator (XPE) is 1.2 watts. The PE evaluation of the adaptive posture consumed 32% of the overall power. The power consumption of other components was also obtained from the XPE analysis. Interfacing with external modules consumes the second-largest share of power.

The total power consumption is 1.2 watts, with the dynamic power and static power consumption comprising 0.96 watts and 0.24 watts. Hardware schemes were designed with eight pipeline stages (S), and a device clock time (Tclk) of 10 ns; a total of 30 (N) iterations were utilized for validation. The 370 ns latency of the proposed hardware schemes is represented in Equations (2) and (3).
Latency per iteration = 8 × 10 ns = 80 ns(2)
Total latency = (N + S − 1) × Tclk = (30 + 8 − 1) × 10 ns = 370 ns.(3)

The overall accuracy is 98.4%. This was computed based on the data captured from the multiple sensors and their fusion. The total number of correct data predictions were 30 and 29, respectively. Equations (4) and (5) represent the accuracy and error rate formulae. An accuracy of 98.4% and error rate of 1.6% are mentioned in Equations (6) and (7).
(4)$$Accuracy=\frac{Number\;of\;Correct\;Predictions}{Total\;Predictions}\times 100$$
Error rate = 1 − Accuracy(5)
(6)$$Accuracy=\frac{29}{30}\times 100=98.4\%$$
(7)$$Error\;rate=1-\frac{29}{30}\times 100=1.6\%$$

### 4.2. Experimental Results

This section describes the experimental setup and experiments for the validation of the proposed sleep posture analysis accelerator.

#### 4.2.1. Experimental Setup

Figure 10 illustrates sleep posture analysis using the contactless approach proposed in this study. As shown by the experimental setup in Figure 13, the side views of A_L and A_R are different for each subject. The results are shown in Figure 14. The experimental setup presents the integration of the six ultrasonic sensor pairs from rhydolabz that utilize an ultrasound echo signal. The sensors were operated with a 40 KHz frequency and voltage range of 3.3–5 V, and consumed a 5 mA current. The voltage was fetched from the voltage regulators of 7805 IC modules. Ultrasonic sensors were employed to detect objects using pulse-width modulation (PWM)-based echo signals that were digitized using an FPGA. The ultrasonic sensor is capable of capturing data within the ideal distance of 3 m; however, to remove redundancy and other noise, a 2.7 m to 0.3 m range was more suitable.

The PIR sensor was positioned between the H_L and H_R sensors to estimate whether the bed was occupied by a human. The PIR sensor range covered around 3 m from its position. As shown in Figure 14, the FPGA device was placed on a bedside table and was used to compute the sleep posture as per the proposed approach. The bed size used in this experiment was approximately 110 cm. The sensor was positioned from the bed at a height of 2.4 m, and each sensor covered approximately 47 cm of the bed. The sensors were positioned to cover an inner-bed range of 94 cm. Figure 10 shows the sensor positions on the ceiling.

#### 4.2.2. Experimental Results of Sleep Posture Learning

The experimental results of the proposed sleep posture learning method are illustrated in Figure 15. Three subjects participated in this experiment. The human postures were evaluated using ultrasonic sensorsIn this regard, the proposed method learns the distance between sensors and determines the subject’s posture without using any database sets. In processing the database sets, huge amounts of computing and processing power are consumed at the inference stage. Figure 15a presents the supine posture of subject 1, which is displayed on the FPGA Zed Board LCD display. The resulting sleep posture details are transmitted through the IoT module Wi-Fi ESP8266 to the monitoring or assisting team for their next course of assistance. Similarly, other postures of subject 1 were learned and registered as a reference for future usage, as in Figure 15c,d. The supine and right yearner postures of subject 2 are demonstrated in Figure 15e,f. Subject 3 was positioned in the left yearner posture and is displayed in Figure 15g,h. Figure 15b,d,h represent the supine, left lateral, and left yearner postures on the FPGA Zed-Board LCD display. This display is useful as interim information when transmitting posture details to the assisting team. The experimental results are presented as follows: https://www.youtube.com/watch?v=6nRHrVYnXTQ accessed on 14 September 2024.

#### 4.2.3. Experimental Results of Adaptive Sleep Posture Analysis

After learning the sleep posture at the inference stage, the proposed approach provides a better solution for the estimation of posture analysis under both generic and adaptive event conditions. Figure 16a–f demonstrate the adaptive sleep postures of the initial subject in the experiment. In Figure 16a, the subject exhibits the left yearner (LY) posture. The subject changed to the right yearner (RY) pose, as shown in Figure 16c, and the supine posture (SP), as shown in Figure 16b. The same adaptive sleep posture taken during the interval of around 9.64 s from the LY end time to the RY start time was presented on the FPGA LCD display as TD (duration).

The same subject switched his adaptive posture from right yearner (RY), as demonstrated in Figure 16c, to the left fetal (LF) position, as shown in Figure 16e, in the supine posture. The duration was recorded as 11.44 s. Figure 16d shows the past FPGA Zed-Board LCD display posture as right yearner (RY) and the current posture as left yearner (LY), along with the time duration. Similarly, 16f represents the adaptive posture of the left yearner to the right foetus. The experimental results are as follows (https://www.youtube.com/watch?v=Z8UvHXnd6lY accessed on 14 September 2024).

Table 4 presents a comparison with other research methods of sleep posture analysis. Most sleep posture detection methods have been developed for use with a bedsheet/mat. The authors of [17,25] used more sensors for accuracy, as any misleading data from sensors could impact the sleep posture analysis. However, this affected the power consumption. R. Tapwal et al. [26] utilized two costly and limited flex force sensors in a method that detected up to four postures only and proportionally consumed more power, at 17.5 watts. Hu, D et al. [27] utilized 32 piezoelectric ceramic sensors for analysis, achieving better results for nuanced pressure disturbances. The proposed methods perform sleep posture analysis in generic and adaptive scenarios, can be carried out using an FPGA-based accelerator with a low power consumption of around 1.2 watts, and can be operated using computing modules with a 100 MHz clock frequency; meanwhile, other comparison methods have a higher power consumption and require CPU resources. Data acquisition was performed using the Arduino Nano or Uno operated at 16 MHz. In the proposed method, a single reconfigurable device provided better results (98.4%) for both data acquisition and analysis.

## 5. Conclusions

Sleep posture analysis has attracted considerable attention as a means of monitoring patients/children and the elderly. The proposed approach is the first of its kind to provide a solution with hardware schemes. Hardware schemes were adopted, alongside machine-learning-based heuristic methods, in the processing of sleep posture analysis at the learning, classification, and evaluation stages with processing elements (PEs). Sound-based data acquisition was successful in concurrently capturing and fusing data at a rate of 25 µs. The proposed method provides a better solution at the inference stage by using hardware schemes with adaptive subject sleep posture recognition and analysis with standard forms. This avoids excessive memory use at the learning and evaluation stages. Each subject and each posture-learning method was validated in 30 iterations, and the latency of the proposed hardware was around 370 ns. The results of the experiment showed 98.4% accuracy and a 1.6% error rate. The resource consumption of the optimized hardware schemes was 51% for the BRAM, 46% for the LUTs, and 44% for the DSP slices. Overall, 1.2 watts of power was consumed for computation. It is hoped that the device will be optimized in the future via partial reconfiguration methods and multi-subject sleep posture detection for hospital patients and senior citizens.

## Figures and Tables

**Figure 1 sensors-24-07104-f001:**
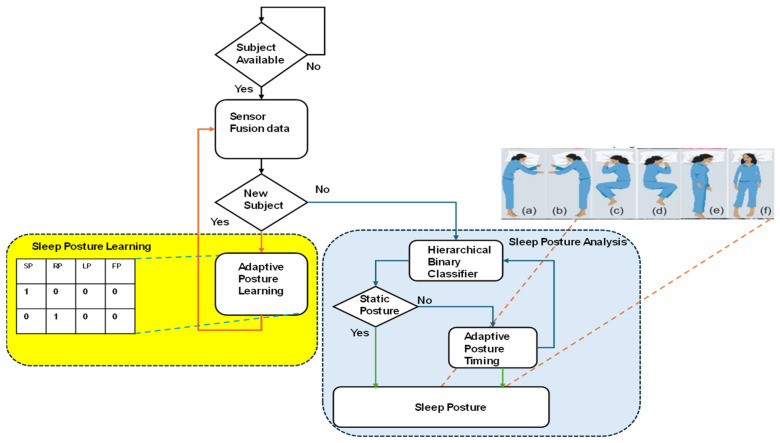
Flowchart of proposed hardware-based sleep posture analysis. (**a**) right yearner (RY), (**b**) left yearner (LY), (**c**) left fetal (LF), (**d**) right fetal (RF), (**e**) left lateral posture (LLP), and (**f**) supine posture (SP).

**Figure 2 sensors-24-07104-f002:**
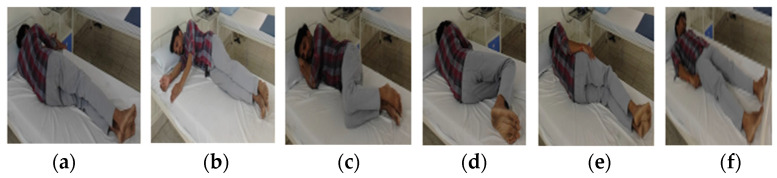
(**a**) right yearner (RY), (**b**) left yearner (LY), (**c**) left fetal (LF), (**d**) right fetal (RF), (**e**) left lateral posture (LLP), and (**f**) supine posture (SP) [17].

**Figure 3 sensors-24-07104-f003:**
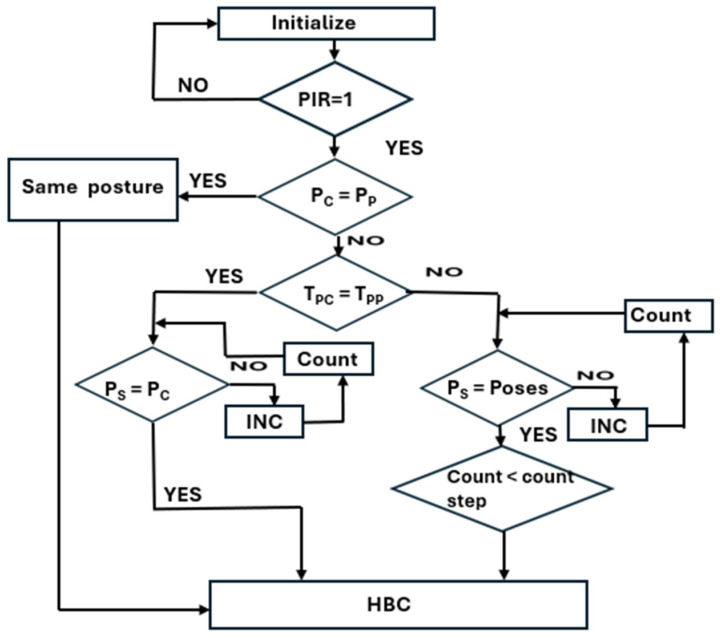
Flowchart for sleep posture learning.

**Figure 4 sensors-24-07104-f004:**
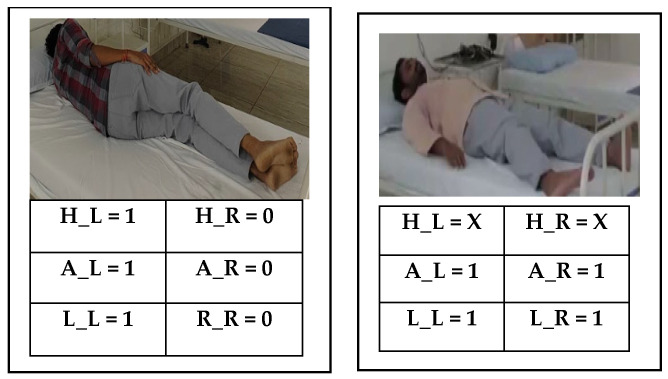
Left lateral posture (LLP) and supine posture (SP) [17] with sensor fusion data.

**Figure 5 sensors-24-07104-f005:**
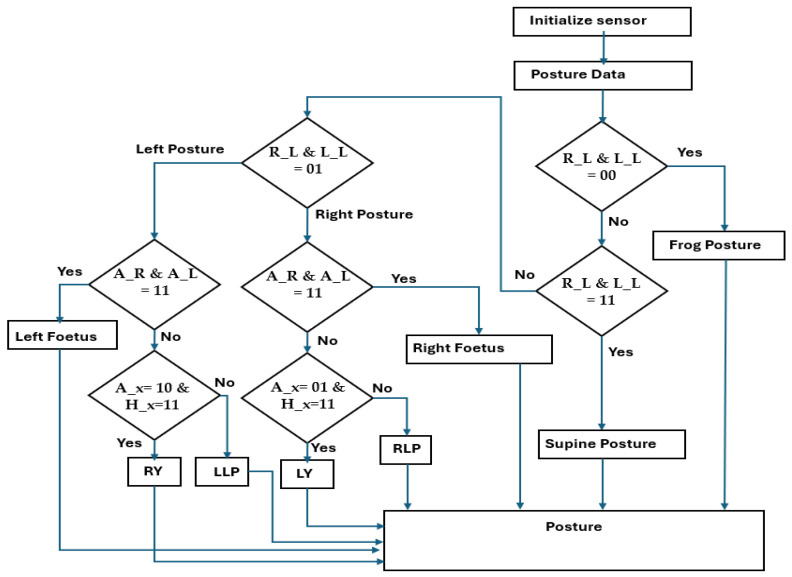
Flow chart of hierarchical binary classifier algorithm for sleep posture.

**Figure 6 sensors-24-07104-f006:**
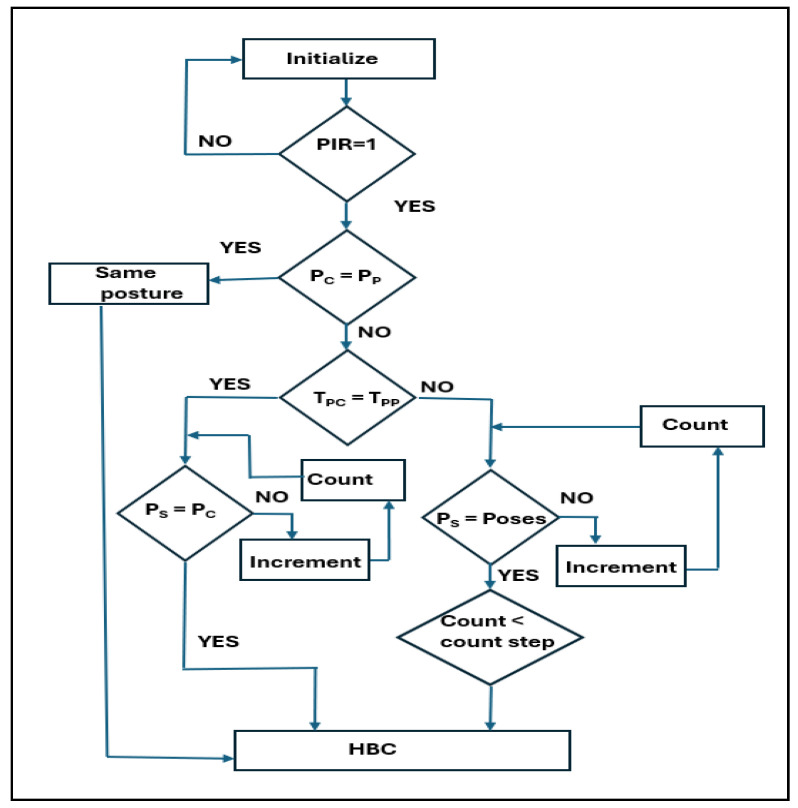
Flow chart of adaptive-based algorithm for sleep posture analysis.

**Figure 7 sensors-24-07104-f007:**
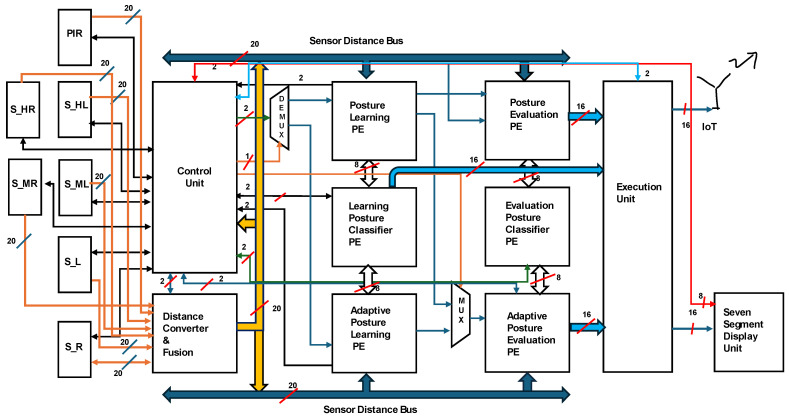
Overall hardware accelerator for sleep posture analysis.

**Figure 8 sensors-24-07104-f008:**
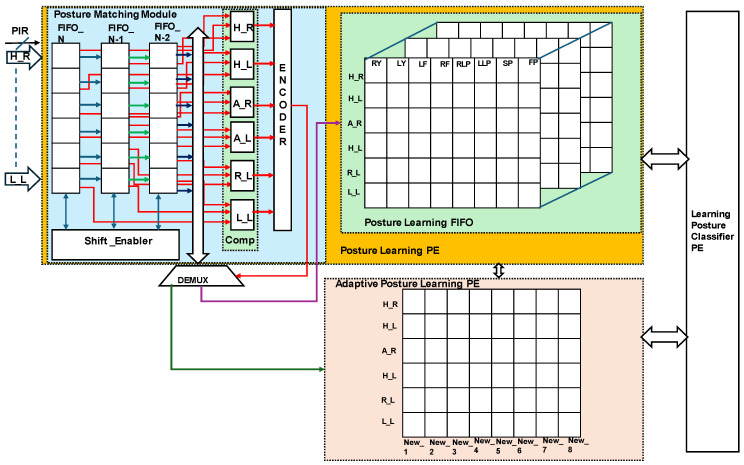
Internal architecture of sleep-posture-based learning.

**Figure 9 sensors-24-07104-f009:**
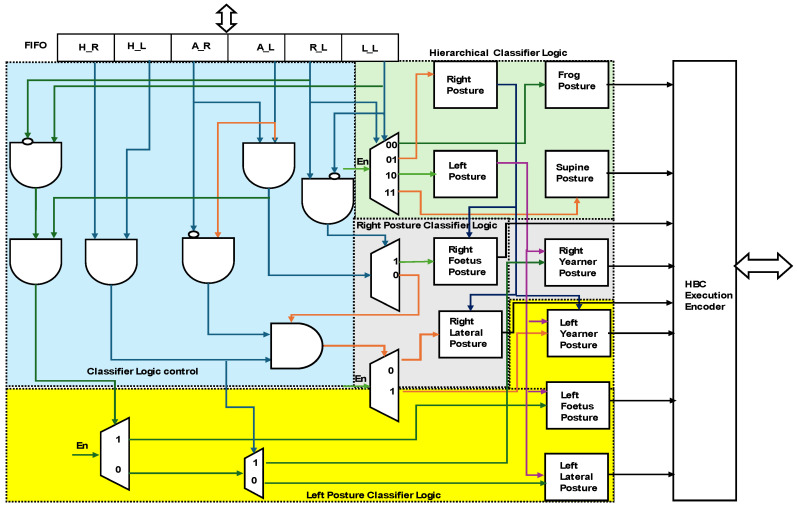
Hardware scheme for sleep-posture-based hierarchical binary classifier (HBC) PE.

**Figure 10 sensors-24-07104-f010:**
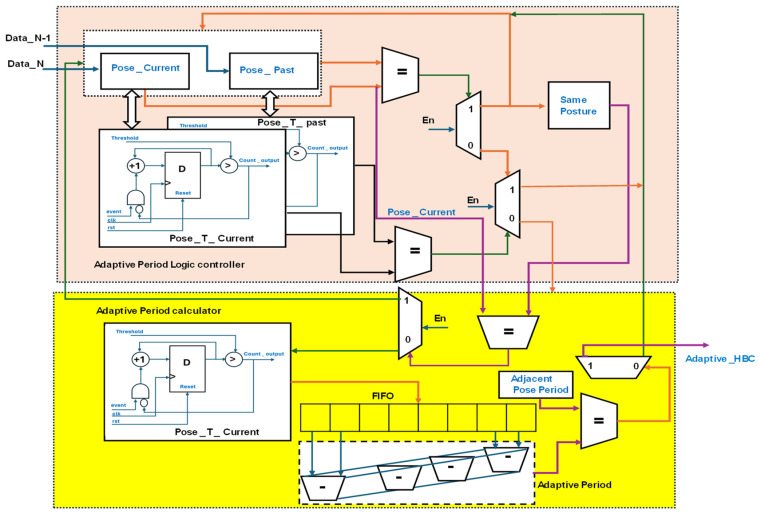
Hardware scheme for adaptive posture evaluation PE.

**Figure 11 sensors-24-07104-f011:**
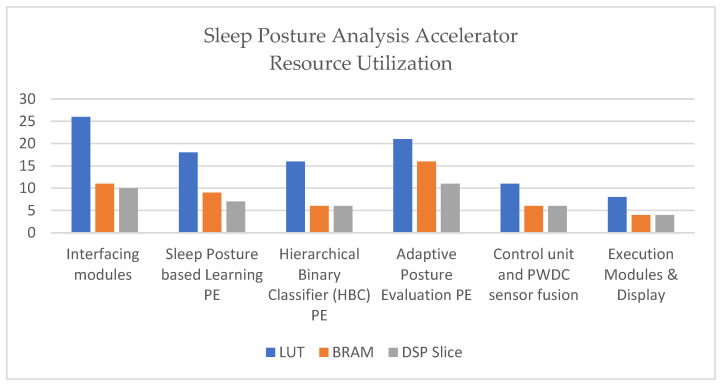
Resource utilization of sleep posture analysis accelerator.

**Figure 12 sensors-24-07104-f012:**
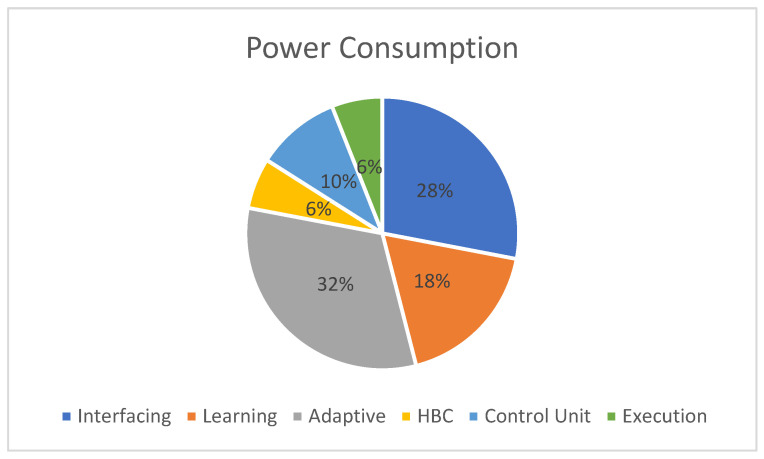
Device power consumption of sleep posture analysis accelerator.

**Figure 13 sensors-24-07104-f013:**
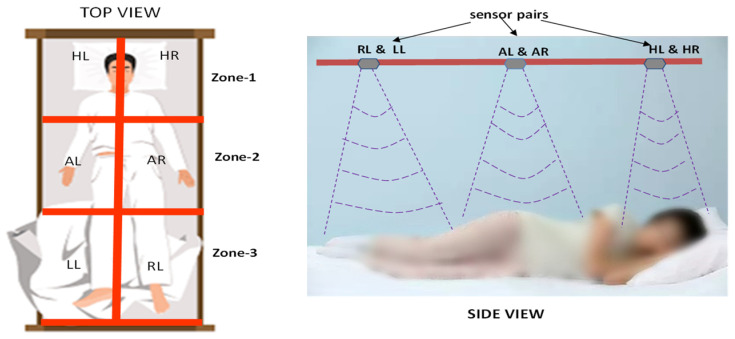
Illustration of experimental setup of contactless sleep posture analysis.

**Figure 14 sensors-24-07104-f014:**
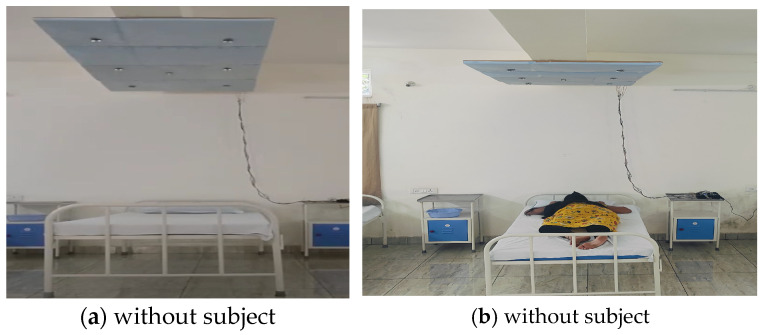
Experimental setup of bed for sleep posture analysis without and with human.

**Figure 15 sensors-24-07104-f015:**
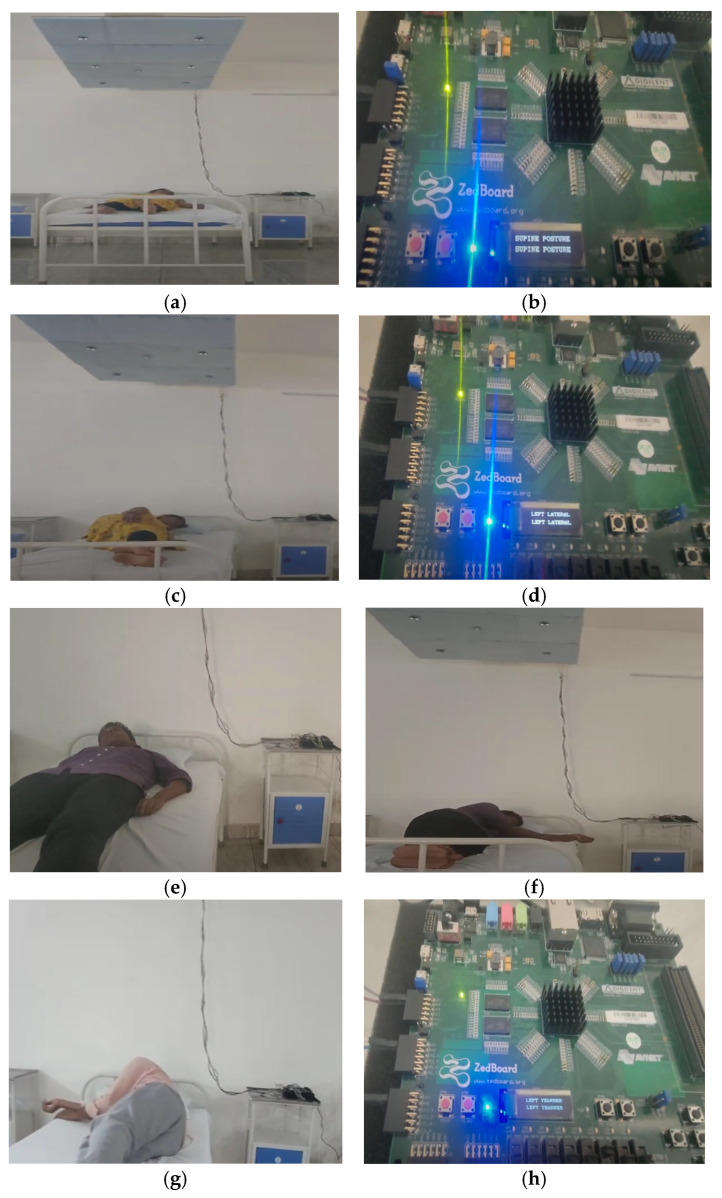
(**a**–**h**) Demonstration of results of sleep posture learning of various subjects. (**a**) Subject_1 Supine Posture. (**b**) Subject_1 Supine Posture Display. (**c**) Subject_1 Left Lateral Posture. (**d**) Subject_1 Left_Lateral_Display. (**e**) Subject_2 Supine Posture. (**f**) Subject_2 Right Yearner. (**g**) Subject_3 Left Yearner. (**h**) Subject_3 Left Yearner display.

**Figure 16 sensors-24-07104-f016:**
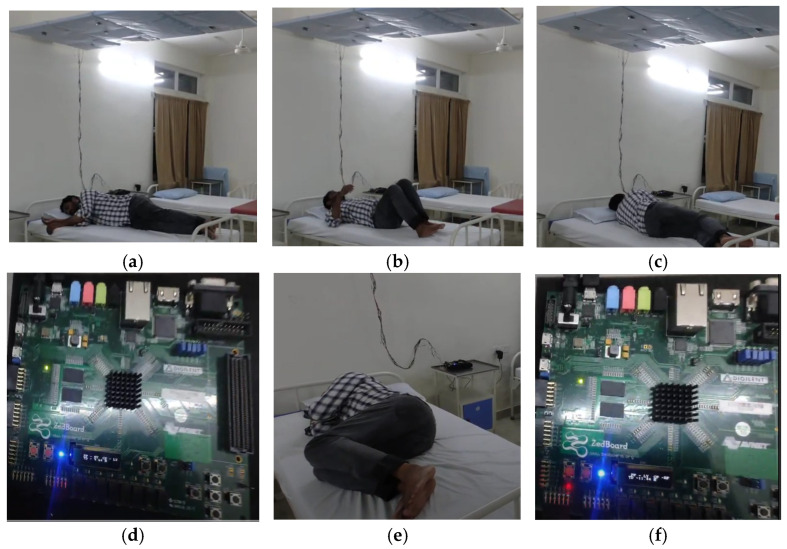
(**a**–**f**) Demonstration of results of adaptive sleep posture analysis.

**Table 1 sensors-24-07104-t001:** Proposed research-related abbreviations.

Abbreviation	Definition
S_PR_	S: Ultrasonic sensors {H_R, H_L, A_R, A_L, R_L and L_L} P: Position of sensor at head (H), abdomen (A), and limb (L) R: Position at right (R) and left (L) sides
PP	Past posture
CP	Current posture
T	Time
Postures	Right yearner (RY), left yearner (LY), left fetal (LF), right fetal (RF), left lateral posture (LLP), and supine posture (SP)
LUT	Look-up table
P_C_	Pose current
Pp	Pose past
Tp_C_	Pose T current
Tpp	Pose T past
Ps	Pose Static

**Table 2 sensors-24-07104-t002:** Sensor fusion data for sleep posture analysis.

Posture	Head_Right (H_R)	Head_Left (H_L)	Abdomen_Right (A_R)	Abdomen_Left (A_L)	Right Leg (R_L)	Left Leg (L_L)
Right yearner (RY)	1	1	0	1	0	1
Left yearner (LY)	1	1	1	0	1	0
Left fetal (LF)	1	1	1	1	0	1
Right fetal (RF)	1	1	1	1	1	0
Right lateral posture (RLP)	1	0	1	0	1	0
Left lateral posture (LLP)	0	1	0	1	0	1
Supine posture (SP)	X	X	1	1	1	1
Frog posture (FP)	1	1	1	1	0	0

**Table 3 sensors-24-07104-t003:** ZedBoard FPGA resource utilization for sleep posture analysis accelerator.

Module	LUT	BRAM	DSP Slices
Interfacing modules (sensors, communication (UART), Xilinx IP cores)	6362	16	22
Sleep-posture-based learning PE	4404	12	15
Hierarchical binary classifier (HBC) PE	3916	8	12
Adaptive posture evaluation PE	5140	22	25
Control unit and PWDC sensor fusion	2692	9	12
Execution modules and display	1958	5	10
Total	24,472	72	96

**Table 4 sensors-24-07104-t004:** Comparison of sleep posture analysis with relevant research methods.

Reference Paper	Sensory Approach		Algorithm	Hardware	Number of Postures	Pros	Accuracy	Cons
	Method	Fusion						
Q. Hu et al., 2021 [17]	1024 pressure sensors	Yes	HOG, SVM, and CNN	Arduino Nano and CPU	6	<400 ms, sampling, and processing,	86.94% to 91.24%	Contact approach
Mater et al., 2020 [25]	1728 FSR sensors	Yes	HOG + LBP, FFANN	CPU	4	Health monitoring	97%	Increased usage of sensors
R. Tapwal et al., 2023 [26]	Two flex force sensors	Yes	K-means	Arduino Uno and CPU	4	Health monitoring	~99.3%	Consumes 17.5 W, contact approach
Hu, D et al., 2024 [27]	32 piezoelectric sensors	Yes	S^3^CNN	N/A	4	Effectively detects nuanced pressure disturbances	93.0%	N/A
Proposed	6 ultrasonic sensors	Yes	HBC, heuristic learning	FPGA	8	Parallel computing, <370 ns, sampling, and computation.	98.4%	PR flow would be preferred in future usage

## Data Availability

Data is available within the article.

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
