# Peer review of "A Field-Programmable Gate Array-Based Adaptive Sleep Posture Analysis Accelerator for Real-Time Monitoring"

_sensors, 2024, doi:10.3390/s24227104_

Round 1
Reviewer 1 Report
Comments and Suggestions for Authors
1. The power and the latency of the hardware platform should be compared and analyzed in the experimental part.
2. The data storage of the sleeping status is sparse. Make some explanation of whether the the data storage can be futher optimized by compressing the data.
Comments on the Quality of English LanguageI think the writing require some minor revisions. There are some title numbers in the text
Author Response
Dear Reviewer thank you for insights, we have provided the modified manuscript along with response to comments.

Reviewer 2 Report
Comments and Suggestions for Authors
The research specifically focuses on developing a system that can assist the elderly and patient attendees by accurately detecting and classifying sleep postures in real-time. The goal is to provide valuable insights for healthcare professionals and caregivers to improve patient care and address potential sleep-related issues.
The abstract mentions that algorithms were coded in Verilog HDL, simulated and synthesized using VIVADO 2017.3 (why using outdated version?), I think this information is redundant, as the results shouldn’t change if for example VHDL would be used.
Since Algorithm's would be more understandable for the reader, I would advise using them instead of pseudo code.
The justification for utilizing FPGA-based technology appears to be inadequately articulated. Wouldn't a straightforward microcontroller suffice in achieving comparable outcomes, while also offering reduced power consumption?
Presenting results in the form of YouTube videos raises concerns; what are the implications if the video is removed from the platform? I would recommend avoiding the provision of information in a scientific paper in that manner.
The figures presented exhibit low image quality, and the purpose of several figures remains unclear (specifically fig 13d, fig 13f, fig 12b, fig 12d, fig 12h). It is advisable to develop more generalized representations for clarity. It is widely understood how a bed appears both when occupied by an individual and when unoccupied. 😊
The experimental setup is quite basic, provide methodology how the experiments were executed, how many times the different postures were measured and so on.
The method of calculating the 98.4% accuracy is not clearly defined, and it is important to understand the factors that contribute to the error rate.
From line 509 it is stated that: “FPGA zed board LCD display and same have been transmitted to monitoring or assisting team through the IoT module Wi-Fi ESP8266.” It’s not clear what is being transmitted via IoT module, or what is the reason behind it.
Conclusions must be improved as they are not supported by the results. From conclusions it‘s not clear what was achieved, 1.2 W static power consumption is probably not the main achievment of this paper.
Author Response
Dear Reviewer, thank you, for insights, we have provided the modified manuscript along with response to comments.

Round 2
Reviewer 1 Report
Comments and Suggestions for Authors
The revised paper has answered my questions.
Author Response
We wish to convey our gratitude to reviewer and editors for providing the comments, which improved quality of our manuscript.
Reviewer 2 Report
Comments and Suggestions for Authors
Thank you for your answers and improving the paper, my last comment would be that I would still advise to write down the results instead placing an image of a development kit.
Author Response
We wish to convey our gratitude to reviewer and editors for providing the comments, which improved quality of our manuscript.
We have updated the manuscript as per the reviewer’s suggestion and have highlighted the corrections at lines 544–546 and 561–564.